# Survivin Overexpression Has a Negative Effect on Feline Calicivirus Infection

**DOI:** 10.3390/v11110996

**Published:** 2019-10-30

**Authors:** Oscar Salvador Barrera-Vázquez, Clotilde Cancio-Lonches, Carlos Emilio Miguel-Rodríguez, Monica Margarita Valdes Pérez, Ana Lorena Gutiérrez-Escolano

**Affiliations:** Departamento de Infectómica y Patogénesis Molecular, Centro de Investigación y de Estudios Avanzados del IPN, México City 07360, Mexico; osbarrera6@gmail.com (O.S.B.-V.); ccancio@cinvestav.mx (C.C.-L.); carlos_emilio_13@hotmail.com (C.E.M.-R.); mmargaritavp@gmail.com (M.M.V.P.)

**Keywords:** FCV, MNV-1, survivin, fJAM-1, conditioned media

## Abstract

It is known that levels of the anti-apoptotic protein survivin are reduced during *Murine norovirus* MNV-1 and *Feline calicivirus* (FCV) infection as part of the apoptosis establishment required for virus release and propagation in the host. Recently, our group has reported that overexpression of survivin causes a reduction of FCV protein synthesis and viral progeny production, suggesting that survivin may affect early steps of the replicative cycle. Using immunofluorescence assays, we observed that overexpression of survivin, resulted in the reduction of FCV infection not only in transfected but also in the neighboring nontransfected CrFK cells, thus suggesting autocrine and paracrine protective effects. Cells treated with the supernatants collected from CrFK cells overexpressing survivin showed a reduction in FCV but not MNV-1 protein production and viral yield, suggesting that FCV binding and/or entry were specifically altered. The reduced ability of FCV to bind to the surface of the cells overexpressing survivin, or treated with the supernatants collected from these cells, correlate with the reduction in the cell surface of the FCV receptor, the feline junctional adhesion molecule (fJAM) 1, while no effect was observed in the cells transfected with the pAm-Cyan vector or in cells treated with the corresponding supernatants. Moreover, the overexpression of survivin affects neither *Vaccinia* virus (VACV) production in CrFK cells nor MNV-1 virus production in RAW 267.4 cells, indicating that the effect is specific for FCV. All of these results taken together indicate that cells that overexpress survivin, or cell treatment with the conditioned medium from these cells, results in the reduction of the fJAM-1 molecule and, therefore, a specific reduction in FCV entry and infection.

## 1. Introduction

*Feline calicivirus* (FCV), a member of the *Vesivirus* genus in the *Caliciviridae* family, is one of the most important and highly prevalent pathogens of cats that cause a range of clinical diseases, from inapparent infections to mild oral and upper respiratory tract disease, and even systemic infections that are frequently fatal [1]. Therefore, FCV is an important health problem in both wild and domestic cats. Additionally, FCV represents one of the best models for the study of calicivirus biology as it is one of the few cultivable members of the family, some reverse genetic systems have been developed for it, and its receptor has been identified [2,3,4].

The functional receptor for FCV to its target cells is the feline junctional adhesion molecule 1 (fJAM-1) [4,5]. This molecule is widely distributed in feline tissues and is localized at cell–cell junctions of epithelial and endothelial cells. FCV attachment to its cell receptor is followed by clathrin-mediated endocytosis and acidification of endosomes [6], allowing for the uncoating of its genome and release into the cytoplasm. The genome is a single-stranded RNA of positive polarity composed of three open reading frames (ORF); ORF1 encodes for 6 nonstructural (NS) proteins, while ORF2 and ORF3 encode for the major and minor structural proteins named VP1 and VP2. Once in the cytoplasm, the genome is translated to produce the nonstructural (NS) proteins which are required for the establishment of the viral factories known as replication complexes (RC) where the RNA-dependent RNA polymerase (RdRp), together with cellular factors, copy the genomic RNA for the synthesis of the negative-strand RNAs that serve as the templates for the production of two types of positive polarity RNAs: 1) the full-length RNAs that become the genomes of the viral progeny and 2) the subgenomic RNA of 2.4 kb that is the major template for translation of the capsid proteins VP1 and VP2. After genome and capsid assembly, viral particles are released from the cells by apoptosis [7].

It is known that FCV as all caliciviruses undergoes the mitochondrial pathway of apoptosis for successful infection, particularly to facilitate the release and spread of the viral progeny into the host. Evidence of the translocation and activation of pro- and anti-apoptotic molecules has been extensively documented, including translocation of Bax protein into the mitochondria, the release of cytochrome C to the cytosol, activation of caspase-9 and -3 during FCV and murine norovirus 1 (MNV-1), and the downregulation of survivin during MNV [8,9,10,11,12]. Recently, our group has reported the translocation of the pro-apoptotic factor Smac/DIABLO (second mitochondria-derived activator of caspases and direct IAP-binding protein with low PI) from the mitochondria to the cytoplasm, with the concomitant downregulation of the anti-apoptotic proteins survivin and XIAP during FCV infection. Moreover, we found that the leader of the capsid protein (LC)—a 124 aa protein produced by processing of the VP1 precursor protein—is responsible for these changes and for induction of apoptosis [12]. We also found that the inhibition of endogenous survivin degradation induced by FCV infection with lactacystin treatment caused a delay in apoptosis progression, reducing the amount of FCV particles released into the supernatant and without affecting virus production, indicating the association between apoptosis and virus release [12,13]. Moreover, FCV infection of cells that overexpress survivin resulted in a reduction of both viral protein levels and viral yield production [12], suggesting that survivin overexpression affects early stages of the FCV replicative cycle, such as binding and/or entry to permissive cells.

In this work, we wanted to further explore how survivin overexpression may affect the early stages of FCV infection. We found that overexpression of survivin has an autocrine and paracrine protective effect against FCV infection, which is not observed for MNV-1 or vaccinia virus (VACV) infections. The reduced ability of FCV to bind to cells overexpressing survivin correlates with a reduction in the presence of fJAM-1 and its cellular receptor on the cell surface.

## 2. Materials and Methods

### 2.1. Cells and Virus Infection

The murine leukemia macrophage-like cell line RAW 267.4, the Crandell Rees feline kidney cell line (CrFK), and Vero cells obtained from American Type Culture Collection (ATCC) (Rockville, MD, USA) were cultured as previously described [14,15]. Infection with MNV-1 and FCV F9 strains (ATCC) (Rockville, MD, USA) was performed as previously described [9,16]. VACV was propagated in Vero cells as previously described [17]. Plaque assays were performed as previously described [18,19].

### 2.2. Plasmids

The plasmids used in this work are described in [12].

### 2.3. Transient Transfections and Plaque Assays

Transient transfections and plaque assays from total and cell- and supernatant-associated fractions were carried out as previously described [12].

### 2.4. Western Blotting Analysis

Mock-infected and FCV infected cells transfected with pAm-Cyan and pAm-Cyan-survivin vectors were washed with PBS, lysed in Laemmli sample buffer, and boiled for 10 min. Supernatants from cells transfected with pAm-Cyan and pAm-Cyan-survivin vectors were centrifuged at 2500× *g* rpm for 10 min at 4 °C and separated from the pellet. The proteins present in the cell extracts were analyzed by SDS-PAGE and transferred to 0.22 μm pore size nitrocellulose membranes (Bio-Rad). The membranes were blocked with 5% skimmed milk for 2 h and incubated overnight at 4 °C with the following antibodies: anti-JAM-1 (Abcam), anti-survivin, (Cell Signaling Technology); anti-nucleolin C23 (H-250) (Santa Cruz Biotechnology, CA, USA), anti-actin (kindly donated by Manuel Hernández, Cinvestav, México), anti-FCV NS6/7 and anti-MNV-1 NS7 (kindly donated by Ian Goodfellow, University of Cambridge, UK). The blots were washed with 0.05% Tris-buffered saline (TBS)–Tween, incubated for 2 h with the appropriate secondary antibodies, and developed using chemiluminescence (PIERCE, IL, USA). Quantification of protein levels was achieved by measuring band intensities in the scanned images using ImageJ software (http:/rsb.info.nih.gov/ij) (MD, USA) and expressed as arbitrary units. The statistical tests were performed using the GraphPad Prism software (CA, USA). One-way analysis of variance (ANOVA) was used. Error bars represent the standard deviation from three independent experiments.

### 2.5. Immunofluorescence Assays

CrFK cells were grown overnight on glass coverslips and infected with FCV at a multiplicity of infection (MOI) of 5 at the indicated times, or at an MOI of 10 for 30 min at 4 °C as indicated. The cells were treated with cytoskeleton buffer for 5 min and permeabilized in 4% formaldehyde solution for 5 min at room temperature (RT). The samples were washed three times with phosphate buffer saline (PBS) for 5 min, blocked with 0.5% gelatin in PBS for 40 min at RT, washed three times with PBS for 5 min, and incubated with the anti-FCV (FCV1-43, Santa Cruz Biotechnology, CA, USA) that recognizes an epitope on the capsid protein or the anti-JAM-1 (ab106114, Abcam, Cambridge, UK), at 4 °C overnight. Samples were washed three times with cold PBS for 5 min and incubated with the appropriate secondary antibodies (Invitrogen, MA, USA) for 1 h at RT. The samples were washed three times with PBS and incubated with 1 mg/mL of 4′6′-diamidino-2-phenylindole (DAPI) for 2 min. The samples were washed six times with PBS and three times with distilled water. The samples were treated with VECTASHIELD liquid mounting media (Vector Laboratories A.C., CA, USA) and analyzed using a Zeiss LSM-700 confocal microscope.

### 2.6. Virus-Binding Assay by Flow Cytometry

Virus-binding assay was performed as previously described [4]. Briefly, 1 × 10^6^ CrFK cells were washed once with cold PBS, incubated with 3 × 10^5^ FCV particles, for 30 min at 4 °C and washed again with cold PBS. Samples were fixed with PBS–4% paraformaldehyde (PFA) for 20 min, blocked with cold PBS–10% horse serum (HS) for 30 min, washed twice with cold PBS and incubated with the anti-FCV antibody (FCV1-43, Santa Cruz Biotechnology) for 1 h at 4 °C. For fJAM-1 expression assay, 1 × 10^6^ CrFK cells were washed once with cold PBS, incubated with 1 µM cytochalasin E (*Aspergillus clavatus*, Sigma C2149) for 30 min at RT, washed with cold PBS and incubated with PBS–5 mM EDTA for 10 min at RT previous to detachment. Cells were washed once with cold PBS, permeabilized or not with 4% PFA–0.5% Triton X100 for 10 min and fixed with PBS–4% PFA for 15 min, blocked with cold PBS–10% HS for 30 min, washed twice with cold PBS and incubated with an anti-JAM-1 antibody (anti-junctional adhesion molecule 1/JAM-A, Abcam) for 30 min at 4 °C. Cell samples were washed three times with cold PBS and incubated with the corresponding secondary antibody (Goat anti-mouse IgG (H + L) Alexa Fluor 594, Invitrogen, MA, USA) in a 1:100 dilution for 1 h at 4 °C for the virus binding assay, or (donkey anti-rabbit IgG (H + L), Alexa Fluor 594, Invitrogen, MA, USA) in a 1:300 dilution for 30 min at 4 °C for fJAM-1 expression assay. The cells were washed three times with PBS and analyzed with a FACSCytoflex (Becton Dickinson, Franklin Lakes, MN, NJ, USA).

## 3. Results

### 3.1. Overexpression of Survivin Has an Autocrine and a Paracrine Effect against FCV Infection

It was previously reported by our group that overexpression of Cyan-survivin (which is 15 times greater than the expression of the endogenous survivin measured by Western blotting) resulted in a delay in the cytopathic effect (CPE) and a reduction in viral particle production caused by FCV infection that correlated with a reduced level of viral proteins [12], suggesting that survivin may have a negative effect in the early steps of the infection. To further explore the effect of the overexpression of survivin in FCV infection, CrFK cells transfected with the pAm-Cyan-survivin vector for 48 h were subsequently infected with FCV at an MOI of 5 for 5 h, and the infection was followed by confocal microscopy using an anti-VP1 antibody (Figure 1). While infection was clearly detected in the cells transfected with pAm-Cyan vector used as a control (Figure 1A, upper panel), a statistically significant reduction of infection was observed in cells transfected with the pAm-Cyan-survivin vector (Figure 1A lower panel). Interestingly, this reduction in FCV infection was also observed in cells that did not show survivin overexpression, suggesting that the overexpression of survivin in CrFK cells not only has an autocrine but also a paracrine effect against FCV infection [12].

### 3.2. Supernatants Collected from CrFK Cells Overexpressing Survivin Had Reduced FCV Infection

Since the overexpression of survivin induced the reduction of FCV infection in an autocrine and paracrine manner, it was possible that a soluble factor could be responsible for the inhibition of FCV infection. Therefore, the conditioned medium collected from cells transfected with both pAm-Cyan or pAm-Cyan-survivin vectors for 48 h were incubated with a CrFK cell monolayer for 24 h, and subsequently infected with FCV at an MOI of 5 for 5 hpi; the viral particle production in cell- and supernatant-associated fractions was then quantified by plaque assay (Figure 2). The conditioned medium from cells overexpressing survivin induced a reduction of approximately 1 log in the virus yield from both cell- and supernatant-associated fractions, supporting the idea that a soluble factor produced in cells transfected with survivin is responsible for the inhibition of FCV infection (Figure 2A,B respectively). In addition, a 70% reduction of the viral NS6/7 protein level was observed in cells treated with the conditioned medium from survivin overexpressing cells (Figure 2C,D). These results suggest that the conditioned medium from CrFK cells that overexpress survivin have a specific cytoprotective effect against FCV infection.

### 3.3. The Protective Effect of the Conditioned Medium from CrFK Cells that Overexpress Survivin on FCV Infection is Specific

In order to determine whether the protective effect of the conditioned medium from cells overexpressing survivin was specific for FCV infection, its effect on VACV—another virus that can infect CrFK cells—was tested. CrFK cells were treated with the conditioned medium from CrFK cells overexpressing, or not, survivin for 24 h, and later infected with VACV or FCV for 5 h, and the virus yield was analyzed by plaque assay (Figure 3). Similar amounts of total VACV particles production were obtained in both conditions (Figure 3A); however, a reduction of approximately 1 log in the amount of FCV particles production from cells that overexpress survivin was observed in comparison to the number of viral particles produced from the cells treated with the control conditioned medium (Figure 3B), suggesting that this effect is specific for FCV infection. On the other hand, to further corroborate if the protective effect of the conditioned medium was specific for FCV infection or could also affect other members of the *Caliciviridae* family, its effect in MNV-1 was tested. Since MNV-1 cannot infect CrFK cells, these assays were performed in the permissive cell line RAW264.7. MNV-1 particle production in the RAW 267.4 treated with the conditioned medium from cells overexpressing or not survivin was similar (Figure 3). The CPE and the amount of NS7 protein and cell- and supernatant-associated virus yield produced by MNV-1 infection in this condition were also similar (Appendix A). These results indicate that the protective paracrine effect of survivin is specific for FCV infection.

### 3.4. Overexpression of Survivin in RAW 264.7 Cells Reduces MNV-1 Release

Since survivin is a molecule that is also downregulated during MNV-1 infection, it was possible that its overexpression could negatively affect viral infection in an autocrine manner. Thus, to analyze this possibility, RAW 264.7 cells were transfected with pAm-Cyan or pAm-Cyan-survivin vectors for 48 h and infected with MNV-1 at an MOI of 5 for 16 h, and viral protein levels as well as viral particles production and release were evaluated by Western blotting and plaque assays, respectively (Appendix A). Total extracts from RAW 264.7 cells either overexpressing or not survivin, showed similar levels of MNV-1 NS7 protein (Appendix A), indicating that viral translation or earlier steps from the MNV-1 replicative cycle were not altered under this condition. Moreover, similar amounts of cell-associated viral yield were obtained from both conditions, suggesting that survivin overexpression does not affect MNV-1 viral particles production (Appendix A). On the other hand, a statistically significant reduction of approximately 1.5 log of the supernatant-associated viral yield was observed from cells that overexpress survivin in comparison to the control conditions (Appendix A). This result is in accordance with previous data indicating that overexpression of survivin causes a delay in apoptosis induction and, consequently, a reduction in MNV-1 release.

### 3.5. The Overexpression of Survivin Interferes with FCV Binding to the Target Cells

Due to the fact that both overexpression of survivin or treatment with conditioned medium collected from these cells cause a specific reduction in FCV protein production and virus yield, it is likely that an early event during infection, such as virus binding or entry, could be altered. Therefore, to determine if the overexpression of survivin affected the binding of FCV particles to the cell surface, a flow cytometry assay was performed (Figure 4). CrFK cells transfected with pAm-Cyan or pAm-Cyan-survivin vectors for 48 h were incubated with FCV at a MOI of 10 for 30 min at 4 °C, and the cell-binding of viral particles was measured by flow cytometry using an anti-VP1 antibody (Figure 4A). Under the condition of survivin overexpression, a reduction of 85% of the VP1 signal was observed in comparison to the cells transfected with pAm-Cyan vector alone (Figure 4A,B), indicating that overexpression of survivin has a negative effect in FCV binding to CrFK cells.

### 3.6. The Amount of fJAM-1 Protein is Reduced in CrFK Cells Overexpressing Survivin and in Cells Treated with Conditioned Medium

Since FCV binding to target cells was reduced in pAm-Cyan-survivin but not in pAm-Cyan transfected cells, we next explored if the subcellular localization and/or the amount of the cellular receptor fJAM-1 on the cell surface was altered as a consequence of survivin overexpression. The presence of fJAM-1 molecule in the surface of CrFK cells overexpressing or not survivin was analyzed by confocal microscopy and flow cytometry (Figure 4C–F). A reduction of the fJAM-1 molecule on the cell surface was observed in nonpermeabilized cells that were transfected with pAm-Cyan-survivin vector for 48 h in comparison to the cells transfected with the pAm-Cyan alone, where fJAM-1 was observed on the cell membrane (Figure 4C,E). In a similar way, the reduction in fJAM-1 molecules on the cell surface was observed in nonpermeabilized cells treated with the conditioned medium from cells transfected with the pAm-Cyan-survivin vector, in comparison to the cells treated with the conditioned medium from cells transfected with the pAm-Cyan vector alone (Appendix A). Moreover, a significant reduction in the total number of fJAM-1 molecules was also observed in permeabilized cells that overexpress survivin in comparison to the cells transfected with the pAm-Cyan vector alone by flow cytometry (Figure 4F) and by Western blotting (Figure 4G,H).

All these results taken together indicate that survivin overexpression causes a reduction of fJAM-1 on the cell surface and as a consequence a reduction of FCV binding and infection.

## 4. Discussion

Caliciviruses induce apoptosis for an efficient viral spread into the host; therefore, many pro- and anti-apoptotic molecules are modulated during the infection such as the downregulation of the protein survivin, which occurs during both MNV and FCV infection. One of the strategies to better understand the mechanisms involved in an efficient viral replication is to study the cellular molecules whose levels are changed during infection. One widely used strategy is to knock down molecules that are required or overexpressed during infection or to overexpress molecules that are downregulated. Since survivin is involved not only in apoptosis but also in the regulation of the cell cycle [20], we speculated that calicivirus replication would be negatively affected in the presence of this molecule. To this regard, our group recently reported that endogenous levels of survivin caused a delay in apoptosis establishment induced by FCV that specifically affected virus exit [12]. Moreover, the overexpression of survivin also affects both MNV-1 and FCV release from the cells, in accordance with the fact that apoptosis allow calicivirus spread into the host [11]. Another interesting finding was that in conditions where survivin was overexpressed, FCV but not MNV-1 protein production and virus yield were reduced, suggesting that an early event during FCV infection, was altered, such as binding and/or entry. The reduction of FCV infection was observed not only in the cells that overexpressed survivin, but also in the neighboring cells that were not transfected, suggesting that a molecule or molecules secreted from the transfected cells could also affect virus infection. The fact that the treatment with the conditioned medium from cells overexpressing survivin caused a reduction of FCV virus yield, but did not affect the production of MNV-1 in its permissive cells RAW264.7 or VACV, a virus that infects CrFK cells, indicates that this effect was specific for FCV infection.

Due to the fact that a specific early event during FCV infection such as binding/entry was altered in cells overexpressing survivin or treated with the conditioned medium from these cells; but the infection with MNV-1, another calicivirus that has a different surface receptor than FCV, was not altered [21,22], suggested that the overexpression of survivin could particularly affect the levels or the subcellular localization of fJAM-1, the FCV functional receptor. To this regard, a reduction of FCV binding to its target cells correlated with a reduction of fJAM-1 on the surface in both cells overexpressing survivin as well as in cells treated with its conditioned medium, and in comparison to the cells transfected with the pAm-Cyan vector or treated with the corresponding conditioned medium, were a reduction in fJAM-1 on the cell surface was not observed. These results suggest that the reduction of fJAM-1 on the surface of cells overexpressing survivin or treated with its conditioned medium is responsible for the reduction of FCV binding and, as a consequence, of the reduced viral protein production and virus yield.

Survivin is highly abundant in cancer cells where, in addition to the cytoplasmic and nuclear localization observed in interphase cells, it is also detected in the mitochondria and on the surface of exosomes that are constitutively secreted (reviewed in [23]). Interestingly, neighboring cancer cells in culture can be coerced to proliferate and evade apoptosis by exosomally delivered survivin, demonstrating its role in cell-to-cell communication [23,24,25]. Therefore, it is possible that the overexpression of survivin in RAW 267.4 and CrFK cells emulates this condition of the high abundance of intracellular survivin observed in cancer cells and, consequently, the delivery of exosomal survivin into the culture media, which may explain the reduction in the FCV infection of cells treated with the supernatants from cells overexpressing survivin. Furthermore, survivin was detected specifically in the conditioned medium from cells overexpressing survivin, and not from transfected cells with the pAm-Cyan vector; whether extracellular survivin is present in exosomes and if it is delivered into the neighboring cells remains to be determined.

It is also possible that extracellular survivin or other factors secreted during its overexpression may change the access of specific molecules involved in the binding/entry of FCV to its target cells, such as its functional receptor, the adhesion molecule fJAM-1. Extracellular expressed survivin can bind to the surface of human peripheral blood leukocytes (PBL) and induce the activation of -chains of 2-integrins and their ligand, the intracellular adhesion molecule (ICAM)-1 [26]. This possibility is in accordance with the fact that RAW 267.4 cells overexpressing survivin or treated with the conditioned medium from these cells did not alter MNV-1 protein production or virus yield, that firstly depend on the virus interaction with the members of the CD3000 family but not with the adhesion molecule fJAM-1 [21] or other surface molecules that have been reported to exhibit a reduced expression, in a context of increased survivin expression such as VE-cadherin, CD44 and CD31 [27,28]. On the other hand, Fernandez et al., in 2014, reported that overexpression of survivin enhances the expression of a considerable number of cancer-related genes by increasing -catenin-Tcf/Lef transcriptional activity, in a PI3K/Akt-dependent manner [29]. Among them, the vascular endothelial growth factor (VEGF) whose transcription, expression, and accumulation in the supernatant from cells overexpressing survivin favors angiogenesis in an in vivo chicken chorioallantoic membrane assay [29], indicating that the conditioned medium from these cells contain factors that can modify the environment of the neighboring cells, as extensively described. Moreover, the overexpression of an exogenous survivin has also a positive feedback on the endogenous survivin, as found by our group. Interestingly, the chemokine (C–C motif) ligand 2 (CCL2), another cancer-related gene activated by PI3K/Akt-*,* -catenin-Tcf/Lef [30], has been involved in the redistribution of JAM-A in endothelial brain cells via internalization involving micropinocytosis [31]; internalized JAM-A is transiently stored in recycling endosomes and then recruited to the apical side of the endothelial cells. Here, we found that CrFK cells overexpressing survivin resulted in the reduction of total fJAM-1, suggesting that the decrease of fJAM-1 in the cell surface is not due to internalization. The mechanism of fJAM-1 reduction remains to be determined.

Besides its role in calicivirus spread, apoptosis is now considered essential for the establishment of the immunopathogenic phenotype since the host innate immune response to infection is modulated by suppressing the translation of induced interferon-stimulated genes (ISGs) [32]. The induction of apoptosis during calicivirus infection in its natural host has been documented [33,34]; however, the downregulation of survivin has only been demonstrated in in vitro experiments [11,12,35]. Whether survivin downregulation also occurs during FCV infection of the oral or respiratory epithelium remains to be determined.

Here, we found that overexpression of survivin as well as treatment of cells with its conditioned medium causes a reduction in FCV binding and entry to its target cells, due to the reduction of JAM-1 from the cell surface. Whether other surface molecules that exhibit a reduced expression in correlation with increased survivin levels [27,28] could also negatively affect FCV binding to these cells remains to be determined.

## Figures and Tables

**Figure 1 viruses-11-00996-f001:**
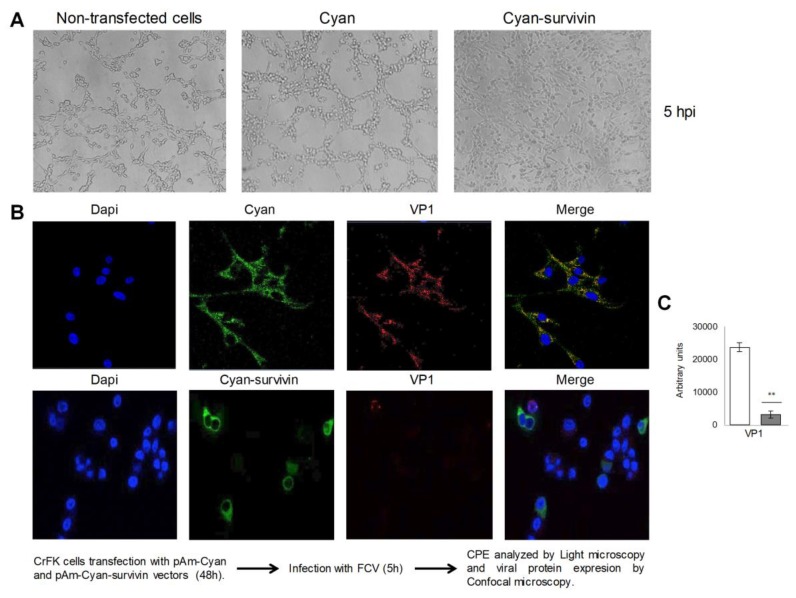
Overexpression of survivin has an autocrine and a paracrine effect against feline calicivirus (FCV) infection. Nontransfected CrFK cells or those transfected with pam-Cyan (Cyan) and pAm-Cyan-survivin (Cyan-survivin) vectors for 48 h were infected with FCV at an MOI of 5 for 5 h. (**A**) The cytopathic effect in nontransfected as well as in transfected CRFK cells was evaluated by light microscopy. Magnification is x40. (**B**) Infection of transfected cells with pam-Cyan (upper panel) and pAm-Cyan-survivin (lower panel) vectors (Green) was determined by immunostaining with an anti-VP1 antibody, followed by Alexa Fluor 594 staining (red). DAPI was used to stain nuclei (blue). The cells were analyzed using a Zeiss LSM 700 laser confocal microscope. Images correspond to a z-stack of 15 slices and are representative of three independent experiments. Merge images are indicated. (**C**) Mean fluorescence intensity of VP1 was determined by Icy software (http://icy.bioimageanalysis.org). ** *p* ≤ 0.001 calculated by two-way ANOVA. Error bars represent the standard deviation from 3 independent assays.

**Figure 2 viruses-11-00996-f002:**
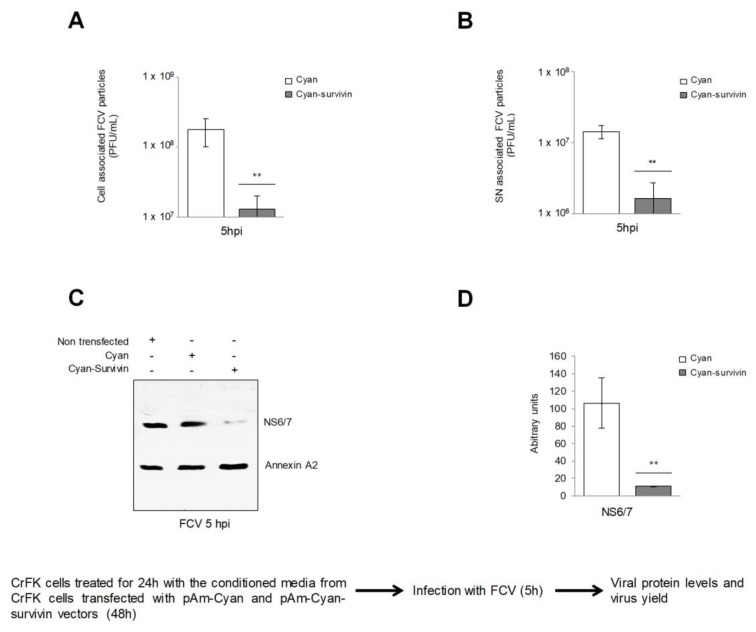
The supernatant from CrFK cells that overexpress survivin reduces FCV infection. CrFK cells were treated for 24 h with the conditioned medium from cells transfected with pAm-Cyan and pAm-Cyan-survivin vectors for 48 h and infected with FCV at an MOI of 5 for 5 h. (**A**) Viral particles production from cell-associated, and (**B**) supernatant-associated fractions were determined by plaque assay. (**C**) Total extracts from CrFK cells transfected with pAm-Cyan and pAm-Cyan-survivin vectors for 48 h and infected with FCV at an MOI of 5 for 5 h were subjected to SDS-PAGE and the levels of NS6/7 protein were analyzed by Western blotting. Annexin A2 was used as the loading control. (**D**) Band intensities of the scanned images were quantified using ImageJ software and expressed as arbitrary units. The statistical tests were performed using GraphPad Prism 7.00 software. ** *p* ≤ 0.001 calculated by two-way ANOVA. Error bars represent the standard deviation from 3 independent assays.

**Figure 3 viruses-11-00996-f003:**
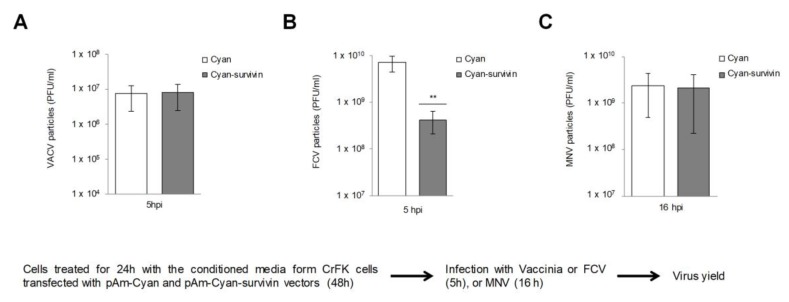
The conditioned medium from CrFK cells that overexpress survivin causes a specific reduction of FCV infection. CrFK cells that are permissive to FCV and vaccinia virus (VACV) infection (**A**,**B**) and RAW 264.7 cells, permissive to MNV-1 infection (**C**) were treated with the conditioned medium from CrFK cells that overexpress (grey) or not (white) survivin for 24 h and infected with (**A**) VACV or (**B**) FCV at an MOI of 5 for 5 h, or (**C**) MNV-1 at an MOI of 5 for 16 h, and virus yields were quantified by plaque assay. The statistical tests were performed using GraphPad Prism 7.00 software. ** *p* ≤ 0.001 calculated by two-way ANOVA. Error bars represent the standard deviation from 3 independent assays.

**Figure 4 viruses-11-00996-f004:**
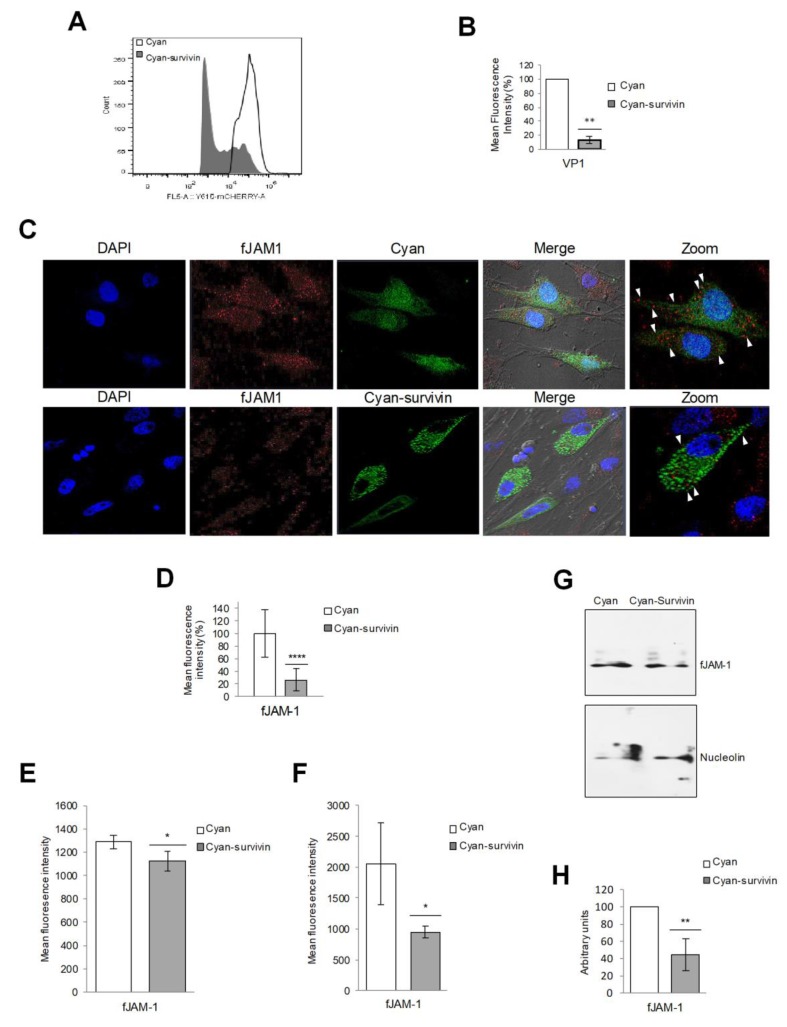
Total and surface fJAM-1 protein levels are reduced in cells overexpressing survivin. (**A**) CrFK cells transfected with pAm-Cyan (Cyan) and pAm-Cyan-survivin (Cyan-survivin) for 48 h were infected with FCV at an MOI of 10 for 30 min at 4°C, immunostained with an anti-VP1 antibody and the amount of bound viruses was analyzed by flow cytometry. Histograms indicate the amount of virus bound in nonpermeabilized cells. (**B**) Mean fluorescence intensity was determined by Icy software. (**C**) Nonpermeabilized CrFK cells transfected with pAm-Cyan and pAm-Cyan-survivin vectors for 48 h were immunostained with an anti-JAM-1 antibody, followed by Alexa Fluor 594 staining (red). DAPI was used to stain nuclei (blue). The cells were analyzed using a Zeiss LSM 700 confocal microscope. The images depict single confocal slices taken from z-stacks. The data shown are representative of at least 3 independent experiments. (**D**) fJAM-1 mean fluorescence intensities were determined by Icy software. (**E**) Nonpermeabilized and (**F**) permeabilized CrFK cells transfected with pAm-Cyan (Cyan) and pAm-Cyan-survivin (Cyan-survivin) for 48 h were treated with cytochalasin E and fJAM-1 expression was measured by flow cytometry. (**G**) Total protein extracts from transfected cells were subjected to SDS-PAGE. fJAM-1 expression was analyzed by Western blotting using a specific anti-JAM-1 antibody. Nucleolin was used as the loading control. (**H**) Band intensities of the scanned images were quantified using ImageJ software and expressed as arbitrary units. The statistical tests were performed using the GraphPad Prism 7.00 software. * *p* < 0.05, ** *p* < 0.01, and **** *p* ≤ 0.001 calculated by two-way ANOVA. Error bars represent the standard deviation from 3 independent assays. White arrows indicate fJAM-1 in the cell surface.

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
