# Peer review of "Survivin Overexpression Has a Negative Effect on Feline Calicivirus Infection"

_viruses, 2019, doi:10.3390/v11110996_

Round 1

Reviewer 1 Report

Manuscript ID: viruses-583859

Title: Survivin Overexpression has a Negative Effect on Feline Calicivirus Infection

General comments:     The objective of the studies described in the present manuscript is to follow up on recently published previous observations that survivin is decreased in feline calicivirus (FCV) infected cells and that over expression of survivin results in reduced virus production.  In this report, the major conclusion is that survivin results in down regulation of the FCV receptor, fJAM, and this results in reduced infection of CrFK cells and thereby a reduction in virus production.

There are several major concerns: 

As an immortalized cell line, CrFK presumably express a high level survivin. The authors should address the biological relevance ‘over expression’. The conclusions would be more impactful if primary cells were also studied. This reviewer is not aware of previous reports that associate survivin function and cell surface receptor down regulation. The literature is vast and perhaps this has been addressed, but the authors should discuss the precedence for this activity if it exists. The conclusions of the study hinge on the demonstration of fJAM down regulation but part of the critical figure (4A) is not fully displayed in the PDF available for review. In addition, whether this is a general phenomenon for all surface protein expression or somehow specific/limited to fJAM should be demonstrated.  Perhaps this can be achieved by showing the impact on other known surface receptors expressed constitutively on CrFK. There are many grammar/syntax issues throughout the manuscript that need to be corrected.

Specific comments: 

Use of ‘form’ versus ‘from’ should be checked throughout.

Line 28: No keywords are provided.

Line 59:  Define Smac/DIABLO

Line 123: Check manufacturer and clarify source.

Figure 1B: Add legend.

Line 170: State the specific statistical test used.

Line 198: Is P<0.5 correct?  What is this referring to?

Line 202: Is there a reference that can be cited for this statement?

Lines 241-259:  The legend needs to be rewritten as it is not understandable.  What is Icy software?  Use and rather than ‘y’.

Line 274: Revise this run-on sentence.

Line 298: Symbols are missing on the PDF submitted for review here and elsewhere.

Line 324: Author contributions are missing.

Author Response

Manuscript ID: viruses-583859

Title: Survivin Overexpression has a Negative Effect on Feline Calicivirus Infection

Reviewer 1.

General comments:     The objective of the studies described in the present manuscript is to follow up on recently published previous observations that survivin is decreased in feline calicivirus (FCV) infected cells and that over expression of survivin results in reduced virus production.  In this report, the major conclusion is that survivin results in down regulation of the FCV receptor, fJAM, and this results in reduced infection of CrFK cells and thereby a reduction in virus production.

There are several major concerns: 

As an immortalized cell line, CrFK presumably express a high level survivin. The authors should address the biological relevance ‘over expression’.

Response: We have compared the amount of survivin in diferent cell lines and we have corroborated that CrFK cells, a cell line from a normal Cat kideney cortex contain less than a half of the amount of survivin observed in the SKOV- 3 cells, an ovarian cancer cell line derived from the ascites of a female with an ovarian serous cystadenocarcinoma, and BEAS-2B cell line, f a lung, bronchus epitelial adenovirus transformed cells. Moreover, transfected cells with the pAM-Cyan-survivin plasmid express aproximatelly 5 times the amount of the endogenous survivin.

The conclusions would be more impactful if primary cells were also studied.

Response: We agree with this reviewer that the observation of this work would have a better impact if primary cells are analyzed, and it is an alternative that, unfortunately, for reasons of time and tools, we can not include it in this work but we will considered for future experiments.

This reviewer is not aware of previous reports that associate survivin function and cell surface receptor down regulation. The literature is vast and perhaps this has been addressed, but the authors should discuss the precedence for this activity if it exists.

Response: Survivin overexpression is associated with reduced expression of cell adhesion molecules such as VE-cadherin, CD44 and CD31, [(Tsuneki, et al, 2014, DOI: 10.1074/jbc.M113.529313. Tsuneki, et al, 2014, DIO: 10.1128/MCB.00671-14.)], in concordance with our finding that the adhesion molecule JAM-1 is reduced from cells that overexpress survivin. This information is now included in the discussion section

Specific comments:

The conclusions of the study hinge on the demonstration of fJAM down regulation but part of the critical figure (4A) is not fully displayed in the PDF available for review.

Response: We apologize for the mistake in the PDF. This figure is now fully displayed in the new manuscript version.

In addition, whether this is a general phenomenon for all surface protein expression or somehow specific/limited to fJAM should be demonstrated.  Perhaps this can be achieved by showing the impact on other known surface receptors expressed constitutively on CrFK.

Response: There are previous works that report the relocation of JAM-1 in response to inflammatory stimuli; for example, when treating the mouse microvascular endothelial cell line (mBMEC) with the chemokine CCL2, a relocation of JAM-1 form the intercellular junctions was induced, towards the interendothelial edge and in a scattered manner, and this phenomenon also occurred with other molecules such as ocludin and claudin 5 [Stamatovic, SM, Sladojevic, N., Keep, RF y Andjelkovic, AV (2012). DOI: 10.1128 / MCB.06678-11]. Moreover, Tsuneki, et al. reported that both CD44KO and CD31KO endothelial cells exhibit a reduced expression of some adhesion molecules such as VE-cadherin, CD44 and CD31 that correlate with an increased survivin expression (Tsuneki, et al, 2014, DOI: 10.1074/jbc.M113.529313). All of the above suggests that the reduction of JAM-1 observed in our work is probably not specific to JAM-1 and might involve the alteration of some other adhesion molecules; thus, it will be interesting in the near future to experimentally corroborate this hypothesis. Information is now included at the end of the discussion section.

There are many grammar/syntax issues throughout the manuscript that need to be corrected.

Response:the grammar/syntax issues were corrected by a native-english spoken person.

Use of ‘form’ versus ‘from’ should be checked throughout.

Response: corrections were inclueded

Line 28: No keywords are provided.

Response: keyword were included.

Line 59:  Define Smac/DIABLO

Response: Definition of Smac/DIABLO is now included

Line 123: Check manufacturer and clarify source.

Response: this information is now included.

Figure 1B: Add legend.

Response: legend was included.

Line 170: State the specific statistical test used.

Response: information was included.

Line 198: Is P<0.5 correct?  What is this referring to?

Response:P value was corrected and missing information included.

Line 202: Is there a reference that can be cited for this statement?

Response:.The way the statement was whritten was confusing; thus, statement was corrected to clarify the idea.

Lines 241-259:  The legend needs to be rewritten as it is not understandable.  What is Icy software?  Use and rather than ‘y’.

Response: Legend was corrected and information regarding Icy is now included.

Line 274: Revise this run-on sentence.

Response: the sentence was corrected

Line 298: Symbols are missing on the PDF submitted for review here and elsewhere.

Response: symbols were corrected

Line 324: Author contributions are missing.

Response: authors contributions were included.

Reviewer 2 Report

Beautifully written and illustrated paper showing that overexpression of survivin prevents feline calicivirus (FCV) from binding to and entering target cells.  

FCV is an ubiquitous infection of domestic and large cats.  Caliciviruses are pathogens of many species on land and in the sea, and due to their ability to recombine, have the potential to create new diseases, as they have done in the past, thus the more we know about them the better. FCV is one of the most studied of the caliciviruses, often used as a surrogate for the human Winter Vomiting and Diarrhoea virus because the latter is difficult to grow in cell culture.

The present study continued the investigations this group has previously made into the replication cycle of FCV and the role of survivin in preventing cell apoptosis, which is necessary for virus transmission to adjacent cells.  I regret that I am not familiar with the survivin literature: please explain in the introduction or discussion how it differs from interferon for readers such as myself, since its action appears similar to that of interferon?  Would interferon have inhibited VACV (but survivin did not)?  Survivin is a protein – has it been further classified as an inteferon, cytokine or something else?  Has it been sequenced?

I would have liked to have seen the cytopathic effects of the FCV-infected cell culture with and without survivin in Figure 1. but that is not necessary for publication – merely a personal preference.

Clearly this is an impressive study which should be published and my comments below are minor criticisms, most of which are simply grammatical or typos, which should however be addressed before publication.  On the whole, the study was very well thought out, with all the appropriate controls put into place.

Line 33: inapparent, not unapparent

Line 49: become not becomes

Line 55: delete several

Line 65: “released into” not associated to

Line 68: affects

Lines 90 & 91: from not form

Line 104: spell out multiplicity of infection (MOI)

Line 105: indicated times – indicated where?

2.5 – please also give the UV microscope make that you used

Line 120: you put horse serum (FBS) – if you mean please spell out what you mean by FBS which usually means fetal bovine serum

Line 123 – Invitrogene or Invitrogen?

Lines 130 &155: particle not particles

Line 130: correlated

Line 150: reduced

Line 258: induce

Lines 274 and 290: replace “To this regard” with Furthermore,

Line 279: suggest not suggests

Line 280: was not is

Line 278: high not highly

Line 292: replace if with whether

Line 295: access to the specific molecules should read access of specific …

Line 296: Delete “To this regard” and begin the sentence with “Extracellular …”

Line 298: is there something missing before chains?

Line 299: accordance with not to

Line 324: Author contributions has nothing written beside it.

Author Response

Reviewer 2.

Beautifully written and illustrated paper showing that overexpression of survivin prevents feline calicivirus (FCV) from binding to and entering target cells.  

FCV is an ubiquitous infection of domestic and large cats.  Caliciviruses are pathogens of many species on land and in the sea, and due to their ability to recombine, have the potential to create new diseases, as they have done in the past, thus the more we know about them the better. FCV is one of the most studied of the caliciviruses, often used as a surrogate for the human Winter Vomiting and Diarrhoea virus because the latter is difficult to grow in cell culture.

The present study continued the investigations this group has previously made into the replication cycle of FCV and the role of survivin in preventing cell apoptosis, which is necessary for virus transmission to adjacent cells.  I regret that I am not familiar with the survivin literature: please explain in the introduction or discussion how it differs from interferon for readers such as myself, since its action appears similar to that of interferon? 

Response: Survivin is an apoptotic and mitotic regulator involved in several cellular pathways and is also involved in the regulation of the expression of adhesion molecules, in concordance with our findings that demonstrated the association among survivin overexpression and the reduction of JAM-1 in the cell surface [(Tsuneki, et al, 2014, DOI: 10.1074/jbc.M113.529313. Tsuneki, et al, 2014, Doi: 10.1128/MCB.00671-14)]. On the contrary, some evidences have show that interferon increases the expression of JAM -1 [(Kobayashi et al, 2008, DOI.org10.1183/09031936.00059507. Kurihara, et al 2013, DOI.org/10.3892/mmr.2013.1419)]. Moreover, it has been reported that cell surface expression levels of JAM-A did not change upon inflammatory stimulation with TNF-α and IFN-γ (Haarmann, et al, 2010 DOI:org/10.1371/journal.pone.0013568 and Marieke et al, 2015, DOI: 10.1161/ATVBAHA.115.305464)]. Thus it is like interferons may regulate FCV infection in a different way than when overexpressing survivin, which may be related to the internalization of its viral receptor.

Would interferon have inhibited VACV (but survivin did not)?  This does not seems to be the case since VACV prevents interferon response (Perdiguero and Esteban. 2009. doi: 10.1089/jir.2009.0073).

Survivin is a protein – has it been further classified as an inteferon, cytokine or something else?  Has it been sequenced?

Response: Survivin has been sequenced and is an anti-apoptotic factor. It has not been clasified as an interferon molecule. Which is interesting is the fact that the increased sensitivity of cancer cells to viruses is due to disruption of innate antiviral defenses associated with dysfunction of type 1 interferons (INFs), (Matveeva et al., 2018. DOI:10.1002/rmv.2008), that correlates with the high amount of survivin.

I would have liked to have seen the cytopathic effects of the FCV-infected cell culture with and without survivin in Figure 1. but that is not necessary for publication – merely a personal preference.

Response: We have included in figure 1 the cytophatic effects of FCV-infected cell culture with and without survivin.

Clearly this is an impressive study which should be published and my comments below are minor criticisms, most of which are simply grammatical or typos, which should however be addressed before publication.  On the whole, the study was very well thought out, with all the appropriate controls put into place.

Line 33: inapparent, not unapparent

Response: correction was inclueded

Line 49: become not becomes

Response: correction was inclueded

Line 55: delete several

Response: correction was inclueded

Line 65: “released into” not associated to

Response: correction was inclueded

Line 68: affects

Response: correction was inclueded

Lines 90 & 91: from not form

Response: corrections were inclueded

Line 104: spell out multiplicity of infection (MOI)

Response: correction was inclueded

Line 105: indicated times – indicated where?

Response: this sentence makes reference to the times in each experiment shown in the figures.

2.5 – please also give the UV microscope make that you used

Response: information was included

Line 120: you put horse serum (FBS) – if you mean please spell out what you mean by FBS which usually means fetal bovine serum

Response: correction was inclueded

Line 123 – Invitrogene or Invitrogen?

Response: correction was inclueded

Lines 130 &155: particle not particles

Response: correction was inclueded

Line 130: correlated

Response: correction was inclueded

Line 150: reduced

Response: correction was inclueded

Line 258: induce

Response: correction was inclueded

Lines 274 and 290: replace “To this regard” with Furthermore,

Response: corrections were inclueded

Line 279: suggest not suggests

Response: correction was inclueded

Line 280: was not is

Line 278: high not highly

Response: correction was inclueded

Line 292: replace if with whether

Response: correction was inclueded

Line 295: access to the specific molecules should read access of specific …

Response: correction was inclueded

Line 296: Delete “To this regard” and begin the sentence with “Extracellular …”

Response: correction was inclueded

Line 298: is there something missing before chains?

Response: the sentence is correct

Line 299: accordance with not to

Response: correction was inclueded

Line 324: Author contributions has nothing written beside it.

This information is now included.

Reviewer 3 Report

This manuscript builds upon authors’ previous work to examine the interaction between survivin expression and Feline Calicivirus infection of CRFK cells.  The work is quite interesting and implicates downregulation the viral receptor as the mechanism for this interaction.  The manuscript is reasonably well written but the veracity of the results can be significantly strengthened by improving the figure legend descriptions to clarify what results are being displayed, and by additional confirmatory tests that would validate the findings. 

Major revisions:

The use of MNV-1 is not explained at all, one assumes this is mouse norovirus 1, but the rationale for using this virus as a control is not provided, and MNV-1 is never defined. The authors indicate that MNV-1 experiments were conducted on a different cell line than CRFK (RAW 267.4 cells) but have not explained how that might confound comparisons, and have not commented about how MNV-1 findings support their conclusions about fJAM-1 downregulation. Figures 1 and 4 are difficult to interpret. The legends do not describe what is what is being measured in each photomicrograph or confocal image, and thus it is very difficult to assess whether the authors’ conclusions are valid.  In particular the fJAM1 images in Figure 4 do not seem to display any binding, though figures E and F do display very low MFI that is statistically different between survivin positive and negative treatments.  Can the authors validate their findings relating to fJAM-1 by conducting flow cytometry or qPCR of fJAM-1 mRNA in the presence/absence of survivin? Can the authors quantify the amount of surviving expression by qPCR (or other method) to verify the magnitude of expression that is required to result in FCV inhibition?

Minor Revisions:

Lines 295-299, Figure 3 legend: Please note that MNV-1 experiments were performed in a different cell line/in vitro system.

Author Response

This manuscript builds upon authors’ previous work to examine the interaction between survivin expression and Feline Calicivirus infection of CRFK cells.  The work is quite interesting and implicates downregulation the viral receptor as the mechanism for this interaction.  The manuscript is reasonably well written but the veracity of the results can be significantly strengthened by improving the figure legend descriptions to clarify what results are being displayed, and by additional confirmatory tests that would validate the findings. 

Major revisions:

The use of MNV-1 is not explained at all, one assumes this is mouse norovirus 1,

Response: In line 69 it was stated that MNV-1 is murine norovirus 1.

but the rationale for using this virus as a control is not provided

Response: the rationale of why we use MNV-1 and Vaccinia is now clarified on lines 229, 236-241.

and MNV-1 is never defined.

Response: MNV-1 is defined on line 70.

The authors indicate that MNV-1 experiments were conducted on a different cell line than CRFK (RAW 267.4 cells) but have not explained how that might confound comparisons,

Response: Information indicating that RAW267.4 but not CrFK is the permissive cell for MNV-1 infection is now included in line 240-242.

and have not commented about how MNV-1 findings support their conclusions about fJAM-1 downregulation.

Response: Since survivin is a molecule that is downregulated by both FCV and MNV-1 infection, we first wanted to determine if the reduced effect in FCV yield due to the overexpression of survivin was specific or has a general effect in calicivirus infection. This is discussed now in lines 237-241. Moreover, we included a comment in the discussion section (lines 364-370), regarding the fact that the effect of the overexpression of survivin affected specifically FCV infection and how this correlated with the fact that infection with MNV-1 that uses another surface receptor for infection was not altered.

Figures 1 and 4 are difficult to interpret. The legends do not describe what is being measured in each photomicrograph or confocal image, and thus it is very difficult to assess whether the authors’ conclusions are valid.  In particular the fJAM1 images in Figure 4 do not seem to display any binding, though figures E and F do display very low MFI that is statistically different between survivin positive and negative treatments. 

Response: Figure legend 1 was corrected and a more detailed description of the results showed is now included. New results are now presented in Fig. 4.

Can the authors validate their findings relating to fJAM-1 by conducting flow cytometry or qPCR of fJAM-1 mRNA in the presence/absence of survivin?

Response: Flow cytometry assays to quantify total fJAM-1 and fJAM-1 in the cell surface of cyan and cyan-survivin expressing cells are now included in figure 4E and 4F. Western blotting of total fJAM-1 protein levels in both cyan and cyan-survivin expressing cells was quantified and also included in figure 4.

Can the authors quantify the amount of surviving expression by qPCR (or other method) to verify the magnitude of expression that is required to result in FCV inhibition?

Response: The levels of the endogenous and transfected survivin proteins in transfected cells were obtained by western blots from at least 3 independent experiments, and quantified using the ImageJ software. This data is now included in lines 165-166.

Minor Revisions:

Lines 295-299, Figure 3 legend: Please note that MNV-1 experiments were performed in a different cell line/in vitro system. 

Response: information is now included in figure legend 3.

Round 2

Reviewer 1 Report

Manuscript ID: viruses-583859

Title: Survivin Overexpression has a Negative Effect on Feline Calicivirus Infection

General comments:     The objective of the studies described in the present manuscript is to follow up on recently published previous observations that survivin is decreased in feline calicivirus (FCV) infected cells and that over expression of survivin results in reduced virus production.  In this report, the major conclusion is that survivin results in down regulation of the FCV receptor, fJAM, and this results in reduced infection of CrFK cells and thereby a reduction in virus production.

Remaining concerns after the first major revision: 

The authors did not address the biological relevance of ‘over expression’ in the paper. The distribution of survivin transcripts in tissues shows very low levels in epithelium of lung and intestine. This is not surprising given the turnover of epithelium that is part of the normal physiology and function in these organs.  Why then would survivin play a role in FCV infection in respiratory epithelium?  Unless the cellular response to FCV infection includes upregulation of survivin as a defense mechanism, the observations described are simply laboratory phenomena. 

To more clearly state the concern: Please include a discussion of the rationale of why survivin might play a role in the pathogenesis of FCV infection in vivo and why overexpression is the appropriate approach to study FCV replication as it relates to natural disease.

The authors nicely describe the importance of apoptosis of FCV infected cells, the induction of pro-apoptotic pathways and the reduced expression of the anti-apoptotic protein, survivin.  Survivin may indeed down-regulate adhesion molecules but how does this play into FCV replication or pathogenesis, in vivo, if the virus naturally infects targets cells with low survivin expression and the host response does not include upregulation of survivin?

Unfortunately the demonstration of fJAM down regulation in figure (4A) is still not fully displayed in the PDF available for review.

Specific comments: 

It does not appear that the legend was not added to Fig. 1B which is now Fig. 1C

Author Response

Response to reviewer

General comments:   The objective of the studies described in the present manuscript is to follow up on recently published previous observations that survivin is decreased in feline calicivirus (FCV) infected cells and that over expression of survivin results in reduced virus production.  In this report, the major conclusion is that survivin results in down regulation of the FCV receptor, fJAM, and this results in reduced infection of CrFK cells and thereby a reduction in virus production.

Remaining concerns after the first major revision: 

The authors did not address the biological relevance of ‘over expression’ in the paper. The distribution of survivin transcripts in tissues shows very low levels in epithelium of lung and intestine. This is not surprising given the turnover of epithelium that is part of the normal physiology and function in these organs.  Why then would survivin play a role in FCV infection in respiratory epithelium?  Unless the cellular response to FCV infection includes upregulation of survivin as a defense mechanism, the observations described are simply laboratory phenomena.

To more clearly state the concern: Please include a discussion of the rationale of why survivin might play a role in the pathogenesis of FCV infection in vivo and why overexpression is the appropriate approach to study FCV replication as it relates to natural disease.

The authors nicely describe the importance of apoptosis of FCV infected cells, the induction of pro-apoptotic pathways and the reduced expression of the anti-apoptotic protein, survivin.  Survivin may indeed down-regulate adhesion molecules but how does this play into FCV replication or pathogenesis, in vivo, if the virus naturally infects targets cells with low survivin expression and the host response does not include upregulation of survivin?

Response: We agree with the reviewer that studding the downregulation of survivin in the FCV natural infection of different tissues would result very interesting. Moreover, in agreement with the reviewer comments, we don´t think that overexpression of survivin would have biological relevance in FCV infection of the respiratory epithelium. Nonetheless, as we have studied and reported that FCV infection down-regulates survivin in cultured cells, we wanted, to further understand the phenomena, and determine if FCV infection could be reduced in the presence of augmented survivin levels. For this purpose, we used two experimental approaches: 1) cells treated with the proteasome inhibitor lactacistin, or 2) survivin overexpression. In both contexts, apoptosis was inhibited, and FCV and MNV-1 exit were reduced, as expected since apoptosis has been mainly associated with virus spread into the host. Therefore, the results obtained under augmented levels of survivin came to reinforce the importance of the apoptosis to favour viral replication. Thus, even though survivin overexpression is purely a laboratory phenomenon, in vitro viral infections are an important tool for that has helped to understand important cellular pathways, in this case, the importance of apoptosis to favour viral replication; of note, the occurrence of apoptosis in the host due to calicivirus infection have been reported. Finally, reporting the finding that overexpressing survivin may affect the presence of fJAM-1 in the cell surface may be useful for others in the field.

We have now modified the introduction (lines 85-91) and included new comments in the discussion section (341-348) that we hope will clarify some of the reviewer's concerns, including that our interest was not to approach the biological relevance of the overexpression of survivin in FCV infection.

Unfortunately, the demonstration of fJAM downregulation in figure (4A) is still not fully displayed in the PDF available for review.

Response: we apologize for this inconvenience, but we checked that the figure was well displayed in the PDF that we included in the reviewed version. fJAM-1 downregulation was also measured by flow cytometry assays and results are now included in new figure 4.

Specific comments: 

It does not appear that the legend was not added to Fig. 1B which is now Fig. 1C

Response: We apologize for this omission, the information is now included in figure legend 1C.

Reviewer 3 Report

The authors have adequately addressed previous concerns and I support publication of this work in Viruses. Sue

Round 3

Reviewer 1 Report

Manuscript ID: viruses-583859

Title: Survivin Overexpression has a Negative Effect on Feline Calicivirus Infection

The authors have made a strong effort to address concerns.  Only minor editing for grammar and syntax is required.